# Lactate Threshold Training Program on Patients with Multiple Sclerosis: A Multidisciplinary Approach

**DOI:** 10.3390/nu13124284

**Published:** 2021-11-27

**Authors:** Alessandra Amato, Paolo Ragonese, Sonia Ingoglia, Gabriella Schiera, Giuseppe Schirò, Carlo Maria Di Liegro, Giuseppe Salemi, Italia Di Liegro, Patrizia Proia

**Affiliations:** 1Department of Psychology, Educational Science and Human Movement, Research Unit, University of Palermo, 90100 Palermo, Italy; alessandra.amato02@unipa.it (A.A.); sonia.ingoglia@unipa.it (S.I.); 2Department of Biomedicine, Neurosciences and Advanced Diagnostics (Bi.N.D.), University of Palermo, 90100 Palermo, Italy; paolo.ragonese@unipa.it (P.R.); schirogiuseppe@libero.it (G.S.); giuseppe.salemi@unipa.it (G.S.); italia.diliegro@unipa.it (I.D.L.); 3Department of Biological, Chemical and Pharmaceutical Sciences and Technologies (STEBICEF), University of Palermo, 90100 Palermo, Italy; gabriella.schiera@unipa.it (G.S.); carlomaria.diliegro@unipa.it (C.M.D.L.)

**Keywords:** multiple sclerosis, physical activity, diet habits, neurotrophins, BDNF, DHEAS

## Abstract

Physical activity could play a key role in improving the quality of life, particularly in patients with nervous system diseases such as multiple sclerosis (MS). Through lactacid anaerobic training, this study aims to investigate the effects at a bio-psycho-physical level to counteract the chronic fatigue associated with the pathology, and to improve mental health at a psychological and neurotrophic level. Eight subjects (age: 34.88 ± 4.45 years) affected by multiple sclerosis were involved. A lactate threshold training program was administered biweekly for 12 weeks at the beginning of the study (T0), at the end of the study (T1) and at 9 months after the end of the study (T2), with physical, psychological and hematochemicals parameters, and dietary habits being tested. The results obtained confirmed that lactacid exercise can influence brain-derived neurotrophic factor (BDNF) levels as well as dehydroepiandrosterone sulfate (DHEAS) levels. In addition, levels of baseline lactate, which could be best used as an energy substrate, showed a decrease after the protocol training. Self-efficacy regarding worries and concerns management significantly increased from T0 to T1. The eating attitudes test (EAT-26) did not highlight any eating disease in the patients with a normal diet enrolled in our study. Physical exercise also greatly influenced the patients psychologically and emotionally, increasing their self-esteem. Lactate threshold training, together with dietary habits, appears to exert synergic positive effects on inflammation, neural plasticity and neuroprotection, producing preventive effects on MS symptoms and progression.

## 1. Introduction

The benefits of physical activity on health are evident in the medical field: prolonged exercise plays a decisive role in modulating certain anti-inflammatory effects [1,2] and can preserve cognitive function during aging [3] and in neuropathological conditions [4]. The relationship between physical activity and the nervous system is evident at various levels. In general, physical exercise is involved in the balance regulation of neurotransmitters [5]. It can improve cognitive and autonomic system functions [6,7] by influencing the concentration of circulating epinephrine and norepinephrine [8] and through modulating the expression of some neurotrophic factors (such as the brain-derived neurotrophic factor (BDNF)), even in pathological conditions [9,10]. Indeed, the BDNF factor is involved in the amplification of the neuroendocrine response, which in turn acts on the growth, survival and maintenance of neurons, both during development and in the adult brain [11,12,13]. In fact, the mechanisms underlying an exercise-induced increase in the BDNF level as a function of intensity are not well known. An important role seems to be played by the increase in neuronal activity evidenced by the increase in c-fos, which is considered to be a marker of neuronal activity, or peNOS (endothelial nitric oxide synthase phosphorylated), which is considered to be a marker of increased cerebral blood flow [14]. Furthermore, exercise causes lactate production, which is also shown to increase the production of growth factors that are important to nervous system function, such as myelination and synaptic plasticity [15,16]. In general, the exercise that determines the increase in lactate production is called lactate threshold (LT) training, which identifies the increase in blood lactate (bL) levels compared with those in the steady state during incremental exercise. It represents a measurement criterion for aerobic endurance performance, and in general, at this level, the production and reuse of lactate are in equilibrium [17]. Another hormone that has neuroprotective effects, improves myelination and has several important biological functions is Dehydroepiandrosterone sulfate (DHEAS), the most abundant steroid hormone in our body. Physical exercise seems to increase the production of this hormone [10]. Moreover, it has been demonstrated that the symptom of fatigue is associated with low levels of DHEAS in subjects with multiple sclerosis [18]. Accordingly, exercise can help to manage fatigue symptoms.

Therefore, physical activity could play a key role in improving the quality of life, particularly in patients with nervous system diseases such as multiple sclerosis (MS). Multiple sclerosis is a chronic demyelinating inflammatory disease that involves the central nervous system (CNS) [19], the etiology of which is considered multifactorial with a clear relevant autoimmune nature. It has been shown that individuals with MS tolerate physical activity, and physical activity can in turn help to manage the symptoms characterizing this pathology, such as: fatigue (not related to effort), walking, balance and coordination problems, visual and cognitive function, numbness, pain and emotional changes [20]. It has been shown that eight weeks of physical training helps to maintain balanced blood–brain permeability in MS patients [21], preventing complications and even playing a neuroprotective role [4]. Furthermore, baseline lactate levels in people with MS appear to be unusually high compared to unaffected subjects [22] as lactate acts as an energy substrate in various cytotypes, in particular in neurons and astrocytes. It is also known that a specific physical exercise, carried out with moderate lactate concentrations, allows us to “train” the organism, in the most efficient way, to reuse the lactate produced [15]. Although the etiology of this pathology is not known, the effects of two fundamental factors are known: genetic and environmental factors. Nutrition, which is one of the environmental factors, plays an important role in both the etiology and comorbid diseases [23].

The severity and duration of the disease, and the level of disability imposed on the individual, cannot be the only factors contributing to psychological adjustments to MS [24]. As outlined by Pakenham and colleagues, the adjustment of individuals to this disease is a complex process in which aspects of their personality, the availability of a support network and the way in which they tend to evaluate their experiences act as potential mediators [25].

A personal disposition which could have an impact on adjustment to MS is self-efficacy, that is the subjective belief that one can overcome challenges that he/she is faced with. In relation to chronic illness, these beliefs affect the individuals’ perception of their condition, the impact of the condition on their life and their capacity for adjustment. Self-efficacy regarding the ability to perform a particular skill or to deal with a specific challenging situation may be considered as a strong predictor of subsequent performance [26,27]. Individuals with a high perception of their self-efficacy will be more likely to set themselves higher aims and be more committed to achieving them than individuals with weak self-efficacy beliefs. The latter, due to their belief that they will not be able to succeed, will tend not to face challenges and to give up easily. Numerous studies have highlighted the influence of self-efficacy on changes in health-related behavioral, including psychological well-being associated with chronic diseases such as MS [28,29]. For example, Wassem (1992) found that among individuals with MS, self-efficacy beliefs regarding self-care, disease management and psychosocial activity correlated with the level of adaptation. In another study with individuals with MS, Barnwell and Kavanagh (1997) found that self-efficacy predicts social activity and the control of negative thoughts, although the most significant overall predictor is past performance. A poor sense of self-efficacy in individuals with MS also seems to be associated with higher levels of depression and the use of dysfunctional coping strategies [30].

Self-determination theory (SDT) [31] has been widely used to study participants’ motivation to exercise [32,33]. SDT postulates the existence of two types of motivation that influences personal behavior: intrinsic motivation (doing a task for the inherent pleasure) and extrinsic motivation (doing an activity for instrumental reasons or to avoid disapproval) [34,35]. The extrinsically motivated behaviors are expressed in four kinds of regulations: external regulation (influenced by external contingencies), introjected regulation (performing to obtain social approval or to avoid internal pressure), identified regulation (recognition and acceptance of the behavior) and integrated regulation (accepting and integrating behavior in other aspects of the self) [31]. These regulatory mechanisms indicate varying degrees of internalization of behavior, reflecting the transition of habits and demands into approved values and self-regulation. This is particularly important in the study of exercise behavior. As this process progresses successfully, individuals can vary between controlled (extrinsic and introjected regulations) and autonomous motivation (identified and integrated regulations) [31]. The latter is a well-internalized extrinsic motivation and, along with intrinsic motivation, is an important factor in continuous exercise adherence [34].

Therefore, the aim of this study was to investigate the effect of a lactate threshold training program, designed to improve the quality of life in subjects with MS, by monitoring motor parameters: strength, balance and eye–hand reaction; psychological parameters: self-efficacy and motivation in exercise; blood chemistry parameters: baseline lactate (investigating the possibility of reusing the lactate itself as an energy substrate), BDNF concentrations and DHEAS levels (considered a putative marker of fatigue in MS); and nutrition habits.

## 2. Methods

### 2.1. Subjects

We recruited patients with diagnoses of MS, confirmed according to McDonald criteria (as revised in 2010 by trained neurologists [36] at the multiple sclerosis outpatients Center of Neurology Unit at the “Paolo Giaccone” University Hospital), according to the inclusion criteria described below:-Aged between 20 and 55;-Absence of clinical relapses in the 12 months preceding the study;-Total Expanded Disability Status Scale (EDSS )score not lower than 1.5 and not higher than 3.5;-Absence of other concomitant diseases (tumors, epilepsy, severe cardiovascular diseases, osteoporosis, etc.);-Signed informed consent by the patient.

A total of 30 patients were assessed for eligibility but only 8 (age: 34.88 ± 4.45 years; height: 168.25 ± 8.66 cm; weight: 72.31 ± 17.28 kg) were included in the study and completed the 12 weeks of the lactate threshold training period. The reasons why patients were excluded, or why they did not complete the study, are described in the recruitment process flow-chart (Figure 1).

### 2.2. Anthropometric Analysis and Coordination Skills Test

The evaluation of anthropometric parameters and body composition was carried out using the Body Fat Analyzer BT-905 bioimpedance meter (Skylark), while the evaluation of coordination skills was carried out using the following five tests:**Timed up and go test**: measures the level of mobility of the subjects and requires static and dynamic balance skills [37].**Eye–hand reaction test**: records the time interval between the presentation of a visual stimulus and the execution of a response.**Flamingo test**: a total body equilibrium test performed without shoes testing the ability of the patient to balance on one leg for 60 s [38].**Wall squat test**: assesses the isometric strength of the lower limbs [39].**Handgrip test**: measures the maximum strength of the muscles of the hand and forearm through the use of a digital dynamometer [40].

All these evaluations were performed before the start of the training period (T0).

### 2.3. Psychological Assessment

**Self-efficacy in patients with multiple sclerosis**. Before and after the training period, the participants were administered the Multiple Sclerosis Self-Efficacy Scale (MSSS) [24]. This consists of 14 items articulated in four subscales: (a) independence and activity; (b) worries and concerns; (c) personal control; and (d) social confidence.

**Motivation to exercise**. Participants were administered the Behavioral Regulation in Exercise Questionnaire-3 (BREQ-3) [41]. It consists of 24 items articulated in five subscales: (a) amotivation; (b) external regulation; (c) introjected regulation; (d) identified regulation; and (e) intrinsic regulation.

**Visual Analogue Fatigue Scale (VAFS).** This test assesses the feeling of fatigue and consists of a vertical line of 10 cm with written descriptions at each end (no fatigue; very intense fatigue). Subjects are asked to mark on the line the point they believe represents their perception of the current state of fatigue. The possible score ranges from 0 to 100, measured in millimeters. The score is obtained by measuring the line from “no fatigue” to the point indicated by the subject, which represents their level of fatigue; the higher the VAFS score, the greater the fatigue [42].

### 2.4. Eating Disorders Evaluations

The eating attitudes test (EAT-26) was used to measure the symptoms and concerns that are characteristic of eating disorders. The eating attitude test was developed by Garner and Garfinkel (1979) [43] and later modified in its present form to the 26-element form by Garner, Olmsted, Bohr and Garfinkel (1982) [44]. It consists of a six-item Likert-type scale with multiple-choice answers. The total score is determined by adding the scores of all the elements, with 20 points considered the threshold above which it is necessary to visit a specialist to investigate an eating disorder.

### 2.5. Hematological Evaluations

Blood samples were collected by a specialist in the Center of Neurology Unit at the “Paolo Giaccone” University Hospital in the morning after overnight fasting. Blood samples were collected in specific tubes containing EDTA for plasma and were centrifuged immediately at 1509× *g* for 10 min at 4 °C. All samples were immediately stored at −80 °C until analyzed.

Biomedical assessments were performed through blood sampling carried out at the beginning of the study (T0), at the end of the study (T1) and after 9 months from the end of the study (T2), and were aimed at the quantitative determination of neurotrophic levels, in particular, BDNF and DHEAS. For the ELISA test, we used a MyBioSource (San Francisco, CA, USA) Biocompare kit and Promega’s GloMax Discover microplate reader.

The determination of lactate levels was also carried out using the Roche Accutrend Plus System portable lactate meter, taking a drop of blood using a lancing device.

### 2.6. Intervention Protocol

The intervention period lasted 12 weeks, within which the load percentage remained unchanged. The protocol was lactate threshold training performed twice a week for a duration of an hour and a half for each training session. It was divided into three phases:Warm up;Workout;Cool down.

The warm up and cool down phases, lasting 15–20 min each, included many proprioceptive, balance and stretching exercises with free body, with the help of mats and barefoot.

The workout phase was characterized by weight exercises carried out in the gym, with the use of specific instruments and with reduced recovery times aimed at producing a modest amount of lactate. Two different training sessions were performed in a week. The exercises and the training characteristics are described in Table 1.

### 2.7. Statistical Analysis

To evaluate the null hypothesis that there is no change between BDNF and DHEAS concentrations in all subjects between T0, T1 and T2, a one-way repeated measured analysis of variance (ANOVA) (SigmaPlot 12, Systat software, San Jose, CA, USA) was conducted, and the Bonferroni correction method was applied to correct for multiple test comparisons with the significance level set to 0.05. Pearson’s correlation was performed to evaluate the correlations between BDNF and DHEAS and between them in T0, T1 and T2. A Student’s t test for paired data was performed to analyze the changes in VAFS score, and the anthropometric and neuromotor abilities between T0 and T1. The results are expressed as mean ± standard deviation. The *p* value was considered significant for values ≤ 0.05. In order to test the differences between the T1 and T0 scores of self-efficacy and motivation to exercise, Wilcoxon tests were performed. SPSS Statistics Version 23 (IBM Corp., Armonk, NY, USA) was used for all statistical tests performed.

## 3. Results

### 3.1. Anthropometric Evaluations

The statistically significant results were highlighted by comparing the values obtained before and after the training period using bioimpedance analysis (BIA), with an increase in total body water (*p* < 0.05) and in basal metabolic rate (*p* < 0.05). Furthermore, an improving trend was detected on the percentages of lean mass (*p* = 0.05) and a tendency to decrease was detected on the percentage of fat mass (*p* = 0.05).

However, the BMI value did not significantly change in either group, measured by comparing the values before and after the 12 weeks of intervention. All results of the anthropometric measurements are shown in Table 2.

### 3.2. Neuromotor Evaluations and Coordination Skills

The results of the MRIs performed before and after the intervention protocol did not show any new lesions that would suggest a possible progression of the disease. This was confirmed by the results of the neurological examination, which highlighted the maintenance of the general health of the subjects, maintaining the same value of EDSS scale from the beginning. Regarding the tests of coordination skills, the “timed up and go test” showed that all subjects achieved results within a normal range (between 11″ and 20″). In the “flamingo test”, there were no statistically significant differences between T0 and T1 (*p* > 0.05). In the “wall squat test”, there were no differences between the right and the left limb at T0 and T1. However, the “wall squat test” score for the left leg significantly increased at T1 (*p* < 0.05), and the right leg had an increased trend score (*p* < 0.05) between T0 and T1. Even in the “handgrip test” (*p* > 0.05) and the “eye–hand reaction test”, there were no differences between T0 and T1. All the results of the anthropometric measurements are shown in Table 2.

### 3.3. Differences in Self-Efficacy and Motivation to Exercise

Results showed significant differences only in the sense of self-efficacy of worries and concerns management (*z* = 2.23, *p* = 0.026, *r* = 0.84), which increased from T0 to T1, and in the regulatory mechanism of introjected regulation (*z* = −2.03, *p* = 0.042, *r* = −0.77), which decreased from T0 to T1.

### 3.4. Eating Disorder Evaluations

The EAT-26 test was used as a screening tool to identify early if patients were suffering from eating disorders. The interviews were carried out with people who scored less than 20 and showed that EAT-26 produces few false negatives. The average score obtained by the subjects enrolled in our project was 10.5 points; this does not exclude the fact that the subjects may suffer from an eating disorder, but does not tend to show an urgency to undergo a diagnostic consultation with a specialist.

### 3.5. Visual Analogue Fatigue Scale

The VAFS results showed a significant score change between T0 and T1 (*p* < 0.05). In particular, the index of reduced fatigue perception showed a decrease from 77.3 ± 13.2 to 58.6 ± 17.3 (Table 2).

### 3.6. Biological Assessment

The results of the ANOVA tests indicated a significant time effect in BDNF concentration (Wilks’ lambda = *p* < 0.05) and in DHEAS concentration (Wilks’ lambda = *p* < 0.01). Thus, there is significant evidence to reject the null hypothesis.

Pairwise comparisons indicated that the BDNF concentration (pg/mL) significantly increased only from T0 to T1 (T0: 351.91 ± 318.04; T1: 612.11 ± 409.48; *p* < 0.05), along with DHEAS concentration (pg/mL) between T0 (8296.24 ± 4182.60) and T1 (10931.26 ± 3698.43; *p* < 0.05) Table 3. Conversely, DHEAS significantly decreased from T0 to T2 (5233.63 ± 2681.97; *p* < 0.05) and from T1 and T2 (*p* < 0.01) (Table 3).

Results from the Pearson’s correlation analysis between BDNF and DHEAS concentrations showed no correlation between T0 (*r* = 0.36), T1 (*r* = 0.14) and T2 (*r* = 0.25) (Figure 2 (A, B, C, respectively)).

Baseline lactate levels showed a decrease after the protocol training between T0 and T1 (*p* < 0.05) (Table 2).

## 4. Discussions

Multiple sclerosis is a multifactorial, inflammatory, autoimmune disease of the central nervous system, with both genetic and environmental components [36,45,46,47]. The precise mechanisms that trigger the pathology are not completely understood; however, it is clear that myelin is specifically targeted, and that, as a result, patients suffer from demyelination, which, in turn, affects the functioning of the nervous system at many levels from the control of movement, to mood, and to awareness of self-efficacy. As a consequence of the broad range of causes and effects involved in the pathology, the studies concerning possible treatments also cover a large range of approaches.

Recently, many studies have reported the beneficial effects of physical activity in a number of neurodegenerative diseases, underlining the neuroprotective effect of exercise [48,49,50,51]. Physical activity can indeed increase the expression of the genes involved in antioxidant defense, can counteract the debilitating effects of many pathologies of the nervous system, and can even improve cognitive functions and memory. As a whole, exercise can at least in part help patients to deal with their daily activities with greater confidence. In particular, it has been reported that MS patients who perform physical exercise have a higher quality of life, experience less fatigue and are less depressed [20]. Exercise can also reduce inflammation in these patients, possibly also through an erythropoietin increase [52]. It was also found that, in MS patients, the interferon-controlled genes are more expressed than in healthy controls; however, after aerobic training, the expression of these genes decreased [53].

In spite of these encouraging results, physical activity is not yet applied as a therapy for MS patients. Thus, the present multidisciplinary study arises from the wish to determine exercise protocols that are able to improve the quality of life of MS patients. In particular, we decided to test the effects of a training protocol performed on the lactacid threshold. In order to evaluate the overall results, we considered psychological, anthropometric, hematochemical and nutritional aspects. The anthropometric measurements showed an improvement in physical composition without, however, a statistically significant change. The only positive data considered the increase in basal metabolic rate (*p* = 0.03). In addition, in the EAT-26 test, which evaluates eating habits, the scores obtained did not highlight any eating disorders; generally, it was recommended that all the patients follow a regular and healthy diet that included a variety of healthy foods from all the food groups.

At the psychological level, by comparing the results obtained before and after the interventions, we observed an increase in the patients’ perception of their own ability to manage concerns related to the disease, and a decrease in the introjected regulatory strategy, i.e., the motivation to exercise only to gain social approval or avoid internal pressure. A limitation of the study is that the research design that was adopted as part of the psychological assessment did not allow for the exclusion of a potential source of bias; that is, that the patients were aware that they were part of a study, and this could have induced them, even unintentionally, to respond in a way that pleased investigators in the post-test phase.

Regarding the concentration of blood lactate (which was, before training, higher than that in average healthy subjects), after the training period, it showed a statistically significant reduction, bringing the values back to the normal range (*p* = 0.01). This finding highlights an improvement in the ability to reuse lactate as an energy substrate, and it is associated with an improvement in the score obtained on the VAS fatigue perception scale (*p* = 0.05).

The most interesting finding concerns the effect of exercise on the levels of BDNF and DHEAS. BDNF is indeed a neurotrophin with a wide range of functions in the nervous system, among which are neuroprotection, the regulation of synaptogenesis and the control of complex cell-to-cell interactions involved in learning and memory [54,55]. On the other hand, neuroactive steroids, such as DHEAS, are important regulators of neuroinflammation, and it has been reported that their levels are reduced in MS patients [56,57]. Moreover, DHEAS can reduce demyelination and axonal loss in the spinal cord [56,57]. On the basis of these considerations, we analyzed the effects of training on the levels of these molecules.

Notably, however, we found that the training protocol had a different effect on the two neurotrophins (BDNF and DHEAS), as evaluated through hematochemical tests: BDNF, indeed, increased significantly (*p* = 0.02) and maintained higher levels, even at the follow-up, carried out after 9 months. On the other hand, DHEAS showed a tendency to increase after the intervention, but without any statistical significance (*p* = 0.05); however, this trend assumes a meaning when it is correlated with the change in the basal metabolic rate. It is known that DHEAS also has an effect on the basal metabolism [58,59]. With this in mind, we suggest that perhaps the duration of the training protocol was not sufficient to highlight a direct effect on the DHEAS levels, but only on the pathways normally influenced by it.

Finally, the analyses performed on BDNF and DHEAS levels at T0, T1 and T2 did not show any correlation, leading us to believe that they follow different and uncorrelated pathways.

Taken together, these findings suggest that lactate threshold training could positively influence the progression of multiple sclerosis, particularly by improving the quality of life. However, given the low number of enrolled patients, we cannot yet generalize our findings. Further studies increasing the sample size and extending the period of intervention will help to clarify whether our results can be extended to all MS patients, and if so, whether novel intervention approaches can be envisaged for future applications.

## Figures and Tables

**Figure 1 nutrients-13-04284-f001:**
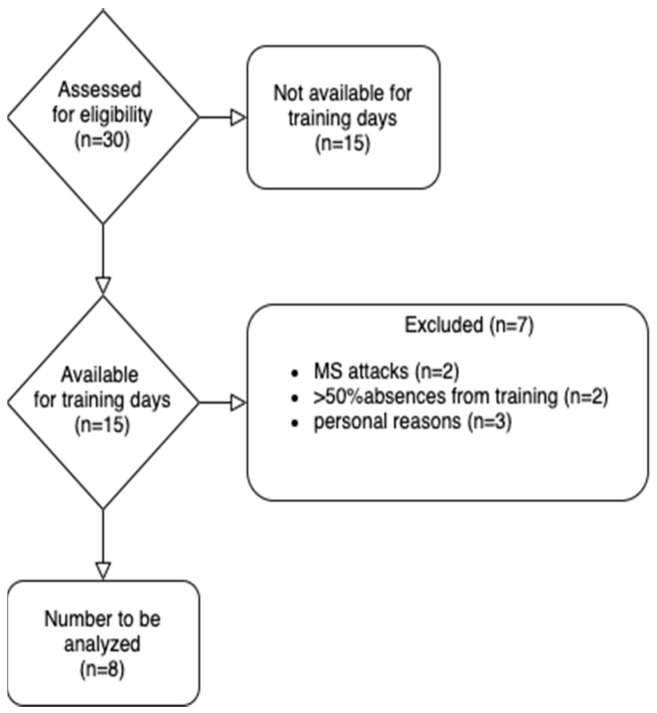
Recruitment process flow-chart.

**Figure 2 nutrients-13-04284-f002:**
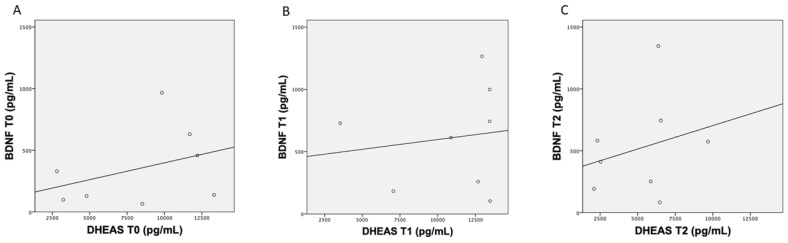
The plots above show the Pearson’s analysis results between brain-derived neurotrophic factor (BDNF) and dehydroepiandrosterone sulfate (DHEAS) concentration at T0 (**A**), between BDNF and DHEAS concentration at T1 (**B**) and between BDNF and DHEAS concentration at T2 (**C**).

**Table 1 nutrients-13-04284-t001:** Description of the characteristics of the exercises performed during the two training sessions of the week. S = set; R = repetitions; RT = repetitions time.

	First Training Sessions	Second Training Sessions
Repetition Time (Minutes)	1.30′	1.30′
1-RM %	50%	50%
Set	2	2
Rest (Minutes)	2–3	2–3
Exercises	Chest press Squat Lat machine Crunches (S = 3; R = 15)	Shoulder press Leg extension Leg curl Plank (S = 3; RT = 30″)

**Table 2 nutrients-13-04284-t002:** Paired Student’s t-test results of anthropometric measurements and physical performance parameters. Data are expressed as mean ± SD. VAFS: visual analogue fatigue scale.

	T0	T1	*p*-Value
Anthropometric measurements			
Weight (Kg)	72.31 ± 17.3	72.18 ± 17.1	0.79
Fat mass (%)	29.31 ± 12,3	21.33 ± 10.5	0.05
Lean mass (%)	70.69 ± 12.3	78.68 ± 10.5	0.05
Heart rate (bpm)	79.38 ± 7.07	79 ± 5	0.92
BMI (Kg/cm^2^)	25.44 ± 5.4	25.39 ± 5.4	0.79
Body water (L)	36.38 ± 7.4	40.9 ± 6.10	0.03 *
Metabolic rate (Cal)	1578.63 ± 210.6	1705.25 ± 184	0.03 *
Physical performance parameters			
Timed up and go (s)	9.08 ± 1.6	10.97 ± 2.4	0.09
Eye–hand reaction (s)	0.50 ± 0.1	0.49 ± 0.1	0.89
Flamingo test (touches)	5.75 ± 8.3	2.13 ± 4.2	0.13
Wall squat R (s)	18.47 ± 11.7	35.39 ± 22.8	0.05
Wall squat L (s)	13.58 ± 8.5	40.86 ± 26.53	0.02 *
Handgrip R (kg)	29.9 ± 16.5	26.24 ± 10.5	0.89
Handgrip L (kg)	26.6 ± 9.1	26.19 ± 8.94	0.37
VAFS	77.3 ± 13.2	58.6 ± 17.3	0.00 *
Basal lactate level (mmol/uL)	2.29 ± 0.4	1.30 ± 0.5	0.01 *

* Significant differences considered for *p* < 0.05.

**Table 3 nutrients-13-04284-t003:** Results of one-way repeated measured analysis of variance (ANOVA), pairwise comparisons.

Time A	Time B	Mean Difference (Time A-B)	SE	Sig. ^a^	95% CI for Difference ^a^	
					Lower Bound	Upper Bound
	BDNF					
T0	T1	−260.200 *	64.289	0.015 *	−461.268	−59.132
	T2	−170.912	126.036	0.652	−565.098	223.273
T1	T2	89.288	123.545	1000	−297.105	475.680
	DHEAS					
T0	T1	−2.632	0.847	0.051	−5.280	0.015
	T2	3.064 *	0.799	0.019 *	0.566	5.562
T1	T2	5.696 *	0.843	0.001 *	3.060	8.332

* The mean difference is significant at *p* < 0.05., ^a^ Adjustment for multiple comparisons: Bonferroni.

## Data Availability

No data related with the review.

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
