# Peer review of "Lactate Threshold Training Program on Patients with Multiple Sclerosis: A Multidisciplinary Approach"

_nutrients, 2021, doi:10.3390/nu13124284_

Round 1

Reviewer 1 Report

This article is quiet clear and provide interesting data.

My concerns are about :

"At the psychological 278 level, comparing the result obtained between pre and post intervention, we observed an 279 increase in patients' perception of their ability to manage concerns related to the disease 280 and a decrease in the introjected regulatory strategy" It could be biased, since the patient is involved in a study. How did you manage this? 

I suggest to authors to provide more details about BDNF PMID  33173466

Author Response

We thank you very much for your comments, suggestions and highlights to improve the quality and clarity of the manuscript. We have revised the manuscript  according with your comments making the changes in the text as you have suggested; over all improving the discussion in the introduction's section about the role of BDNF and inserting a new reference as you suggested.

With regards to comments about the psychological assessment, we understand the reviewer's concerns about a potential bias due to the fact that patients were aware of being part of a study and that this may have led them, even unintentionally, to respond in a way that pleased investigators in the post-test phase. The research design adopted, which envisages that each participant acts as a control of him/herself, does not allow us to exclude the intervention of this potential bias. it is also true that the study adopted a complex assessment strategy (which included, in addition to the psychological one, also the assessment of physical and hematochemicals parameters and dietary habits). From this point of view, the results obtained from the psychological assessment are substantially consistent with those obtained in the other areas tested. Accordingly, we added some sentences in the discussion section.

The changes were highlighted in yellow

Reviewer 2 Report

The present manuscript shows the effects of 12 weeks lactate threshold training on some parameters associated with multiple sclerosis. Although the number of patients included in the study is pretty low, the results might be of interest for the audience to read and to take into consideration for future studies. I consider that the manuscript could be considered after minor revision:

  • A definition of lactate threshold training program should be provided in the introduction.
  • Table 3: I deducted that FU means T2. The authors should be consistent throughout the text, the table and the figure legend when naming this time point.
  • The discussion seems mainly a summary of the results obtained instead of a real discussion. The results should be discussed and compared with other studies.
  • The limitations of the study, such as the low number of patients included, should be stated in the discussion.

Author Response

We thank you very much for your comments, suggestions and highlights to improve the quality and clarity of the manuscript. We have revised the manuscript  making the changes in the text as you have suggested.

First of all we added in the introduction's section the lactate threshold definition according to the Faude et al, 2009.

We corrected the table 3 changing FU with T2,as indicated in the entire paper

We completely rewritten the discussion's section try to better highlighted our finding and the possible application, discussing also about the sample size.

The changes were highlighted in yellow